# A Principled Permutation Invariant Approach to Mean-Field Multi-Agent Reinforcement Learning

## Abstract

Multi-agent reinforcement learning (MARL) becomes more challenging in the presence of more agents, as the capacity of the joint state and action spaces grows exponentially in the number of agents. To address such a challenge of scale, we identify a class of cooperative MARL problems with permutation invariance, and formulate it as mean-field Markov decision processes (MDP). To exploit the permutation invariance therein, we propose the mean-field proximal policy optimization (MF-PPO) algorithm, at the core of which is a permutation- invariant actor-critic neural architecture. We prove that MF-PPO attains the globally optimal policy at a sublinear rate of convergence. Moreover, its sample complexity is independent of the number of agents. We validate the theoretical advantages of MF-PPO with numerical experiments in the multi-agent particle environment (MPE). In particular, we show that the inductive bias introduced by the permutation-invariant neural architecture enables MF-PPO to outperform existing competitors with a smaller number of model parameters, which is the key to its generalization performance.

## 1 Introduction

Multi-Agent Reinforcement Learning (Littman, 1994; Zhang et al., 2019) generalizes Reinforcement Learning (Sutton and Barto, 2018) to address the sequential decision-making problem of multiple agents maximizing their individual long term rewards while interacting with each other in a common environment. With breakthroughs in deep learning, MARL algorithms equipped with deep neural networks have seen significant empirical successes in various domains, including simulated autonomous driving (Shalev-Shwartz et al., 2016), multi-agent robotic control (Matarić, 1997; Kober et al., 2013), and E-sports (Vinyals et al., 2019).

Despite tremendous successes, MARL is notoriously hard to scale to the many-agent setting, as the size of the state-action space grows exponentially with respect to the number of agents. This phenomenon is recently described as the curse of many agents (Menda et al., 2018). To tackle this challenge, we focus on *cooperative MARL*, where agents work together to maximize their team reward (Panait and Luke, 2005). We identify and exploit a key property of cooperative MARL with homogeneous agents, namely the *invariance with respect to the permutation of agents*. Such permutation invariance can be found in many real-world scenarios with homogeneous agents, such as distributed control of multiple autonomous vehicles and team sports (Cao et al., 2013; Kalyanakrishnan et al., 2006), but also in scenarios with heterogeneous agent groups, where invariance holds within each group (Liu et al., 2019b). More importantly, we find that permutation invariance has significant practical implications, as the optimal value functions remain invariant when permuting the joint state-action pairs. Such an observation strongly advocates a permutation invariant design for learning, which helps reduce the effective search space of the policy/value functions from exponential dependence on the number of agents to polynomial dependence.

Several empirical methods have been proposed to incorporate permutation invariance into solving MARL problems. Liu et al. (2019b) implement a permutation invariant critic based on Graph Convolutional Network (GCN) (Kipf and Welling, 2017). Sunehag et al. (2017) propose value decomposition, which together with parameter sharing, leads to a joint critic network that is permutation invariant over agents. While these methods are based on heuristics, we are the first to provide theoretical principles for introducing permutation invariance as an inductive bias for learning value functions and policies in homogeneous systems. In addition, we adopt the DeepSet (Zaheer et al., 2017)

architecture, which is well suited for handling homogeneity of agents, with much simpler operations to induce permutation invariance and greater parameter efficiency.

To scale MARL algorithms in the presence of a large number, even infinitely many, agents, mean-field approximation has been explored to directly model the population behavior of the agents. Mean-field game considers large populations of rational agents that play a noncooperative game. Yang et al. (2017) consider a mean-field game with deterministic linear state transitions, and show that it can be reformulated as a mean-field MDP, where the mean-field state lies in finite-dimensional probability simplex. Yang et al. (2018) take a mean-field approximation over actions, such that the interaction for any given agent and the population is approximated by the interaction between the agent's action and the averaged actions of its neighboring agents. However, the motivation for averaging over local actions remains unclear, and it generally requires a sparse graph over agents. In practice, properly identifying such structure also demands extensive prior knowledge. Mean-field control instead considers a central controller who aims to compute strategy to optimize the average payoff across the population. Carmona et al. (2019) motivate a mean-field MDP from the perspective of mean-field control. The mean-field state therein lies in a probability simplex and is thus continuous in nature. To enable the ensuing Q-learning algorithm, discretization of the joint state-action space is necessary. In addition, the dynamic programming principles of such mean-field control problem has been studied in (Gu et al., 2019). Gu et al. (2020) also propose a Q-learning type algorithm, where the state-action space is first discretized into an epsilon-net. The kernel regression operator is used to construct an estimate of the unknown Q-function from samples. Gu et al. (2021) propose a localized training, decentralized execution framework by locally grouping homogenous agents using their states. Wang et al. (2020) motivate a mean-field MDP from permutation invariance, but assume a central controller coordinating the actions of all the agents, and hence is restricted to handling the curse of many agents from the exponential blowup of the joint state space. Our formulation of mean-field approximation allows agents to make their own local actions without resorting to a centralized controller.

We propose a mean-field Markov decision process motivated from the permutation invariance structure of cooperative MARL, which can be viewed as a natural limit of finite-agent MDP by taking the number of agents to infinity. Such a mean-field MDP generalizes traditional MDP, with each state representing a distribution over the state space of a single agent. The mean-field MDP provides us a tractable formulation to model MDP with many agents, including an infinite number of agents. We further propose the Mean-Field Proximal Policy Optimization (MF-PPO) algorithm, at the core of which is a pair of permutation invariant actor and critic neural networks. These networks are implemented based on DeepSet (Zaheer et al., 2017), which uses convolutional type operations to induce permutation invariance over the set of inputs. We show that with sufficiently many agents, MF-PPO converges to the optimal policy of the mean-field MDP with a sublinear sample complexity independent of the number of agents. To support our theory, we conduct numerical experiments on the benchmark multi-agent particle environment (MPE) and show that our proposed method requires a smaller number of model parameters and attains better performance than multiple baselines.

**Notations**. W denote $\mathcal{P}(X)$ as the set of distribution on set $X$. $\delta_x$ denotes the Dirac measure supported at $x$. For $\mathbf{s} = (s_1, \ldots, s_N)$, we use $\mathbf{s} \overset{\text{i.i.d.}}{\sim} p$ to denote that each $s_i$ is independently sampled from distribution $p$. For $f : X \to \mathbb{R}$ and a distribution $\pi \in \mathcal{P}(X)$, we write $\langle f, \pi \rangle = \mathbb{E}_{a \sim \pi} f(a)$. We write $[m]$ in short for $\{1, \ldots, m\}$, and $\Delta_d$ for the standard probability simplex in $\mathbb{R}^d$.

## 2 PROBLEM SETUP

We focus on studying multi-agent systems with cooperative, homogeneous agents, where the agents within the system are of similar nature and hence cannot be distinguished from each other. Specifically, we consider a discrete time control problem with $N$ agents, formulated as a Markov decision process $(\mathcal{S}^N, \mathcal{A}^N, \mathbb{P}, \mathrm{r})$. We define the joint state space $\mathcal{S}^N$ to be the Cartesian product of the finite state space $\mathcal{S}$ for each agent, and similarly define the joint action space $\mathcal{A}^N$. The homogeneous nature of the system is reflected in the transition kernel $\mathbb{P}$ and the shared reward $\mathrm{r}$, which satisfies:

$$\mathrm{r}(\mathbf{s}_t, \mathbf{a}_t) = \mathrm{r}\left(\kappa(\mathbf{s}_t), \kappa(\mathbf{a}_t)\right), \quad \mathbb{P}(\mathbf{s}_{t+1}|\mathbf{s}_t, \mathbf{a}_t) = \mathbb{P}\left(\kappa(\mathbf{s}_{t+1})|\kappa(\mathbf{s}_t), \kappa(\mathbf{a}_t)\right) \tag{2.1}$$

for all $(\mathbf{s}_t, \mathbf{a}_t) \in \mathcal{S}^N \times \mathcal{A}^N$ and the permutation mapping $\kappa(\cdot) \in \mathbb{S}_N$, where $\mathbb{S}_N$ is the set of all one-to-one mapping from $[N]$ to itself. In other words, it is the configuration, rather than individual identities, that affects the team reward, and the transition to the next configuration solely depends on the current configuration. See Figure 1 for detailed illustration. Such permutation invariance finds applications in many real-world scenarios, including distributed control of autonomous vehicles, and social economic systems (Zheng et al., 2020; Cao et al., 2013; Kalyanakrishnan et al., 2006).

Our goal is to find the optimal policy $\nu$, where $\nu(\mathbf{s}) \in \Delta_{|\mathcal{A}^N|}$ for all $\mathbf{s} \in \mathcal{S}^N$, and maximize the expected discounted reward $V^\nu(\mathbf{s}) = (1 - \gamma)\mathbb{E}\left\{\sum_{t=0}^\infty \gamma^t \mathrm{r}(\mathbf{s}_t, \mathbf{a}_t)|\mathbf{s}_0 = \mathbf{s}, \mathbf{a}_t \sim \nu(\mathbf{s}_t), \forall t \geq 0\right\}$. Our first result shows that learning with permutation invariance advocates invariant network design.

**Proposition 2.1.** *For cooperative MARL satisfying* (2.1)*, there exists an optimal policy $\nu^*$ that is permutation invariant, i.e., $\nu^*(\mathbf{s}, \mathbf{a}) = \nu^*(\kappa(\mathbf{s}), \kappa(\mathbf{a}))$ for any permutation mapping $\kappa(\cdot)$. In addition, for any permutation invariant policy $\nu$, the value function $V(\cdot)$ and the state-action value function $Q(\cdot)$ is also permutation invariant, i.e., $V^\nu(\mathbf{s}) = V^\nu(\kappa(\mathbf{s})),\ Q^\nu(\mathbf{s}, \mathbf{a}) = Q^\nu(\kappa(\mathbf{s}), \kappa(\mathbf{a})),$ where $Q^\nu(\mathbf{s}, \mathbf{a}) = \mathbb{E}_{\mathbf{s}'}\{r(\mathbf{s}, \mathbf{a}) + \gamma V^\nu(\mathbf{s}')\}$.*

Proposition 2.1 has an important implication for architecture design, as it states that it suffices to search within the permutation invariant policy and value function classes. To the best of our knowledge, this is the first theoretical justification of permutation invariant network design for learning with homogeneous agents.

We focus on the factorized policy class with a parameter sharing scheme, where each agent makes its own decision without consolidating with others. Specifically, the joint policy $\nu$ can be factorized as $\nu(\mathbf{a}|\mathbf{s}) = \prod_{i=1}^N \mu(a_i|o_i)$, where $\mu(\cdot)$ denotes the shared local mapping and $o_i$ denotes the local observation. Such a policy class is widely adopted in the celebrated centralized training – decentralized execution paradigm (Lowe et al., 2017), due to its light overhead in the deployment phase and favorable performances. However, directly learning such factorized policy remains challenging, as each agent needs to estimate its state-action value function, denoted as $Q^\nu(\mathbf{s}, \mathbf{a})$. The search space during learning is $(|\mathcal{S}| \times |\mathcal{A}|)^N$, scaling exponentially with respect to the number of agents. The large search space poses as a significant roadblock for efficient learning, and is coined as the *curse of many agents*.

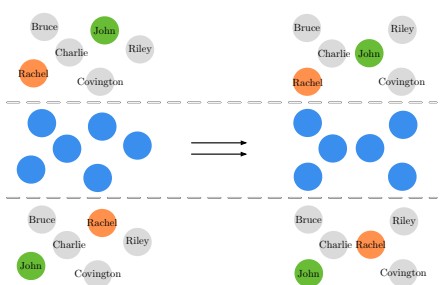

Figure 1: Illustration of permutation invariance. Exchanging identities of the agents (first and third row) does not change the transition of the system (second row).

To address the curse of many agents, we exploit the homogeneity of the system and take the mean-field approximation. We begin by taking the perspective of agent $i$, which is arbitrarily chosen from the $N$ agents. We denote its state as $s$ and the states of the rest of the agents by $\mathbf{s}_r$. One can verify that when permuting the state of all the other agents, the value function remains unchanged; additionally, we can further characterize the value function as a function of the local state and the empirical state distribution over the rest of agents.

**Proposition 2.2.** *For any permutation mapping $\kappa(\cdot)$, the value function satisfies $V^\nu(s, \mathbf{s}_r) = V^\nu(s, \kappa(\mathbf{s}_r))$. Additionally, there exists $g_\nu$ such that: $V^\nu(s, \mathbf{s}_r) = g_\nu(s, \widehat{\mathrm{p}}_{\mathbf{s}_r})$, where $\widehat{\mathrm{p}}_{\mathbf{s}_r} = \frac{1}{N}\sum_{s \in \mathbf{s}_r}\delta_s$ is the empirical distribution over the states of rest of the agents $\mathbf{s}_r$.*

For a system with a large number of agents (e.g., financial markets, social networks), the empirical state distribution can be seen as the concrete realization of the underlying population distribution of the agents. Motivated from this observation and Proposition 2.2, we formulate the following mean-field MDP that can be seen as the limit of finite-agent MDP in the presence of infinitely many homogeneous agents.

**Definition 2.1** (mean-field MDP)**.** *The mean-field MDP consists of elements of the following: state $(s, \mathrm{d}_\mathcal{S}) \in \mathcal{S} \times \mathcal{P}(\mathcal{S})$; action $\overline{a} \in \overline{\mathcal{A}} \subseteq \mathcal{A}^\mathcal{S}$; reward $\mathrm{r}(s, \mathrm{d}_\mathcal{S}, \overline{a})$; transition kernel $\mathbb{P}(s', \mathrm{d}'_\mathcal{S}|s, \mathrm{d}_\mathcal{S}, \overline{a})$.*

The mean-field MDP has an intimate connection with our previously discussed finite-agent MDP. Since the the agents are homogeneous, the system is the same from any agent's perspective. We choose any agent (referred to as representative agent), the state information of such an agent includes the local state $s$, and the mean-field state $\mathrm{d}_\mathcal{S}$. With state information, the agent selects a meta action $\overline{a} \in \overline{\mathcal{A}} \in \mathcal{A}^\mathcal{S}$, and uses such a meta action to make local decision $a = \overline{a}(s) \in \mathcal{A}$. We remark that such a modeling of decision process allows the agent to make decision on both its local information (local state $s$) and the global information (mean-field state $\mathrm{d}_\mathcal{S}$). From homogeneity we assume all the rest of the agents uses the same meta action $\overline{a}$ to make their local actions. Note that different agents can still make different local actions due to their different local states, i.e., $\overline{a}(z) \neq \overline{a}(z')$ in general for

$z \neq z' \in \mathcal{S}$. The joint state at the next timestep $(s', d'_{\mathcal{S}})$ naturally depends on the current global state $(s, d_{\mathcal{S}})$ and the meta action $\overline{a}$ (since all the other agents use $\overline{a}$ to generate their local actions), and is specified by the transition kernel $P(s', d'_{\mathcal{S}} | s, d_{\mathcal{S}}, \overline{a})$. In addition, the representative agents receives a reward $r(s, d_{\mathcal{S}}, \overline{a})$, which depends on the local state and mean-field sate, and the meta action $\overline{a}$.

Our goal is to learn efficiently a policy $\pi$, where $\pi(\cdot | s, d_{\mathcal{S}}) \in \Delta_{|\overline{\mathcal{A}}|}$ for all $(s, d_{\mathcal{S}}) \in S \times \mathcal{P}(\mathcal{S})$, for maximized expected discounted reward. To facilitate discussions, we define the value function $V^\pi(s, d_{\mathcal{S}}) = (1 - \gamma)\mathbb{E}\left\{\sum_{t=0}^\infty \gamma^t r(s_t, d_{\mathcal{S},t}, \overline{a}_t)\right\}$, where $(s_0, d_{\mathcal{S},0}) = (s, d_{\mathcal{S}}), \overline{a}_t \sim \pi(s_t, d_{\mathcal{S},t}), \forall t \geq 0$; and Q-function $Q^\pi(s, d_{\mathcal{S}}, \overline{a}) = (1 - \gamma)\mathbb{E}\left\{\sum_{t=0}^\infty \gamma^t r(s_t, d_{\mathcal{S},t}, \overline{a}_t)\right\}$, where $(s_0, d_{\mathcal{S},0}) = (s, d_{\mathcal{S}}), \overline{a}_0 = a, \overline{a}_t \sim \pi(s_t, d_{\mathcal{S},t})$. The optimal policy is denoted by $\pi^* \in \arg\max V^\pi(s, d_{\mathcal{S}})$.

Despite the intuitive analogy to finite-agent MDP, solving the mean-field MDP poses some unique challenges. In addition to having an unknown transition kernel and reward, the mean-field MDP takes a distribution as its state, which we do not have complete information of during training. In the following section, we propose our mean-field Neural Proximal Policy Optimization (MF-PPO) algorithm that, with a careful architecture design, can solve such mean-field MDP in a model-free fashion efficiently.

## 3 MEAN-FIELD PROXIMAL POLICY OPTIMIZATION

Our algorithm falls into the category of the actor-critic learning paradigm, consisting of alternating iterations of policy evaluation and improvement. The unique features of MF-PPO lie in the facts: (1) it exploits permutation invariance of the system, reducing search space of the actor/critic networks drastically and enables much more efficient learning; (2) it can handle a varying number of agents. For simplicity of exposition, we consider a fixed number of agents here.

Throughout the rest of the section, we assume that for any joint state $(s, d_{\mathcal{S}}) \in \mathcal{S} \times \mathcal{P}(\mathcal{S})$, the agent has access to $N$ i.i.d. samples $\{s_i\}_{i=1}^N$ from $d_{\mathcal{S}}$. We denote concatenation of such samples as $\mathbf{s} \in \mathcal{S}^N$ and write $\mathbf{s} \overset{\text{i.i.d.}}{\sim} d_{\mathcal{S}}$. MF-PPO maintains a pair of actor (denoted by $F^A$) and critic networks (denoted by $F^Q$), and uses the actor network to induce an energy-based policy $\pi(\overline{a} | s, d_{\mathcal{S}})$. Specifically, given state $(s, d_{\mathcal{S}})$, the actor network induces a distribution on set $\overline{\mathcal{A}}$ according to $\pi(\overline{a} | s, d_{\mathcal{S}}) \propto \exp\left\{\tau^{-1} F^A(s, d_{\mathcal{S}}, \overline{a})\right\}$, where $\tau$ denotes the temperature parameter. We use $\pi \propto \exp\left\{F^A\right\}$ to denote the dependency of the policy on the energy function.

• **Permutation-invariant Actor and Critic**. We adopt a permutation invariant design of the actor and critic network. Specifically, given individual state $s \in \mathcal{S}$ and sampled states $\mathbf{s} \in \mathcal{S}^N$, the actor (resp. critic) network $F^A$ (resp. $F^Q$) satisfies $F^A(s, \mathbf{s}, \overline{a}) = F^A(s, \kappa(\mathbf{s}), \overline{a})$ for any permutation mapping $\kappa$. With permutation invariance, the search space of the actor/critic network polynomially depends on the number of agents $N$.

**Proposition 3.1.** *The search space of a permutation invariance actor (critic) network is at the order of* $\left(\sum_{k=1}^{\min\{|\mathcal{S}|, N\}} \binom{N-1}{k-1}\binom{|\mathcal{S}|}{k}\right) |\mathcal{S}||\overline{\mathcal{A}}|$; *Additionally, if $|\mathcal{S}| < N$, then the search space depends on $N$ at the order of $N^{|\mathcal{S}|}$.*

Compared to architectures without permutation invariance, whose search space depends on $N$ at the order of $(|\mathcal{S}||\mathcal{A}|)^N$, we can clearly see the search space of MF-PPO is exponentially smaller.

Motivated by the characterization of the permutation invariant set function in Zaheer et al. (2017), the actor/critic network in MF-PPO takes the form of Deep Sets architecture, i.e., $F^A(s, \mathbf{s}, \overline{a}) = h\left(\sum_{s' \in \mathbf{s}} \phi(s, s', \overline{a})/N\right)$. Both networks first aggregate local information by averaging over the output of a shared sub-network among agents, before feeding the aggregated information into a subsequent network $h(\cdot)$. See Figure 2 for detailed illustration. Effectively, by the average pooling layer and the preceding parameter

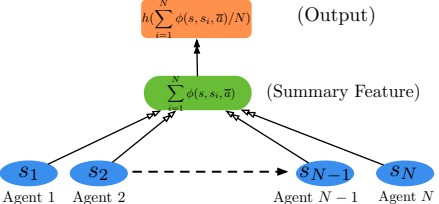

Figure 2: Illustration of a DeepSet parameterized critic network.

sharing scheme, the network can keep its output unchanged when permuting the ordering of agents. Compared to a Graph Convolutional Neural Network (Kipf and Welling, 2017), which uses two sets of weights for the linear transformation layer, one for the the agent itself and one for the aggregation state coming from the rest of the agents. The averaging operation is well suited for homogeneous agents and more parameter-efficient. It also naturally allows us to handle varying number of agents during training and evaluation, which is not readily achievable by GCN network.

Naturally, the actor network when given the joint state-action pair $(s, \mathrm{d}_\mathcal{S}, \overline{a})$ is given by $F^A(s, \mathrm{d}_\mathcal{S}, \overline{a}) = h\left(\mathbb{E}_{s' \sim \mathrm{d}_\mathcal{S}} \phi(s, s', \overline{a})\right)$. We assume $F^A$ is parameterized by a neural network with parameters $\alpha \in \mathbb{R}^D$, which is to be learned during training. We let the function class of all possible actor networks be denoted by $\mathcal{F}^A$. This same architecture design applies to the critic network, with learnable parameters denoted by $\theta \in \mathbb{R}^D$ and the function class denoted by $\mathcal{F}^Q$. MF-PPO then consists of successive iterations of policy evaluation and policy improvement described below.

● **Policy Evaluation**. At the $k$-th iteration of MF-PPO, we first update the critic network $F^Q$ by minimizing the mean squared Bellman error while holding the actor network $F^{A,k}$ fixed. We denote the policy induced by the actor network as $\pi_k$, the stationary state distribution of policy $\pi_k$ as $\nu_k$, and the stationary state-action distribution as $\sigma_k(s, \mathrm{d}_\mathcal{S}, \overline{a}) := \nu_k(s, \mathrm{d}_\mathcal{S}) \pi_k(\overline{a}|s, \mathrm{d}_\mathcal{S})$. Thus, we follow the update

$$\theta_k = \operatorname{argmin}_{\theta \in \mathbb{B}(\theta_0, R_\theta)} \mathbb{E}_{\sigma_k} \left\{ F_\theta^Q(s, \mathrm{d}_\mathcal{S}, \overline{a}) - \left[ \mathcal{T}^{\pi_k} F_\theta^Q \right](s, \mathrm{d}_\mathcal{S}, \overline{a}) \right\}^2, \tag{3.1}$$

where $\mathbb{B}(\theta_0, R_\theta)$ denotes the Euclidean ball with radius $R_\theta$ centered at the initialized parameter $\theta_0$, and the Bellman evaluation operator $\mathcal{T}^\pi$ is given by $\mathcal{T}^\pi \mathcal{F}_\theta^Q(s, \mathrm{d}_\mathcal{S}, \overline{a}) = \mathbb{E}\left\{ (1-\gamma) r(s, \mathrm{d}_\mathcal{S}, \overline{a}) + \gamma \mathcal{F}_\theta^Q(s', \mathrm{d}_\mathcal{S}', \overline{a}') \right\}$, where $(s', \mathrm{d}_\mathcal{S}') \sim \mathbb{P}(\cdot|s, \mathrm{d}_\mathcal{S}, \overline{a}), \overline{a}' \sim \pi(\cdot|s', \mathrm{d}_\mathcal{S}')$. We solve (3.1) by T-step temporal-difference (TD) update and output $\theta_k = \theta(T)$. At the $t$-th iteration of the TD update,

$$\theta(t + 1/2) = \theta(t) - \eta\left\{ F_{\theta(t)}^Q(s, \mathbf{s}, \overline{a}) - (1-\gamma) r(s, \mathrm{d}_\mathcal{S}, \overline{a}) - \gamma F_{\theta(t)}^Q(s', \mathbf{s}', \overline{a}') \right\} \nabla_\theta F_{\theta(t)}^Q(s, \mathbf{s}, \overline{a}),$$
$$\theta(t+1) = \Pi_{\mathbb{B}(\theta_0, R_\theta)}\left(\theta(t+1/2)\right),$$

where we sample $(s, \mathrm{d}_\mathcal{S}, \overline{a}) \sim \sigma_k$, $(s', \mathrm{d}_\mathcal{S}') \sim \mathbb{P}(\cdot|s, \mathrm{d}_\mathcal{S}, \overline{a})$, $\overline{a}' \sim \pi_k(\cdot|s', \mathrm{d}_\mathcal{S}')$, $\mathbf{s} \overset{\text{i.i.d.}}{\sim} \mathrm{d}_\mathcal{S}$, $\mathbf{s}' \overset{\text{i.i.d.}}{\sim} \mathrm{d}_\mathcal{S}'$. We use $\Pi_X(\cdot)$ to denote the orthogonal projection onto set X, and $\eta$ to denote the step size. Note that here for the simplicity of analyses, we sample the state-action pair $(s, \mathrm{d}_\mathcal{S}, \overline{a})$ independently from the stationary distribution. We remark that trajectory samples can also be used, which essentially requires bounding the bias of the gradient at each iteration due to the dependencies between trajectory samples, and we can readily apply the fast-mixing property of Markov chains to control such a bias (Bhandari et al., 2018). The details of policy evaluation are summarized in Algorithm 2, Appendix A.

● **Policy Improvement**. Following the policy evaluation step, MF-PPO updates its policy by updating the policy network $F^A$, which is the energy function associated with the policy. We update the policy network by

$$\pi_{k+1} = \underset{\substack{\pi \propto \exp\{F^{A,k+1}\}, \\ F^{A,k+1} \in \mathcal{F}^A}}{\operatorname{argmax}} \mathbb{E}_{\nu_k} \left\{ \left\langle F_{\theta_k}^Q(s, \mathrm{d}_\mathcal{S}, \cdot), \pi(\cdot|s, \mathrm{d}_\mathcal{S}) \right\rangle - \upsilon_k \mathrm{KL}\left(\pi(\cdot|s, \mathrm{d}_\mathcal{S}) \| \pi_k(\cdot|s, \mathrm{d}_\mathcal{S})\right) \right\}.$$

The update rule intuitively reads as increasing the probability for choosing action $\overline{a}$ if it yields a higher value for critic network $F^Q(s, \mathrm{d}_\mathcal{S}, \overline{a})$, which can be viewed as a softened version of policy iteration (Bertsekas, 2011). Additionally, by controlling the proximity parameter $\upsilon_k$, we can control the softness of the update, with $\upsilon \to 0$ yielding the vanilla policy iteration. Moreover, without constraint of $F^{A,k+1} \in \mathcal{F}^A$, such an update would have a nice closed form expression, and $\pi_{k+1}$ itself is another energy-based policy.

**Proposition 3.2.** *Let $\pi_k \propto \exp\left(\tau_k^{-1} F_{\alpha_k}^A\right)$ denote the energy-based policy, then the update*

$$\overline{\pi}_{k+1} = \operatorname{argmax}_\pi \mathbb{E}_{\nu_k} \left\{ \left\langle F_{\theta_k}^Q(s, \mathrm{d}_\mathcal{S}, \cdot), \pi(\cdot|s, \mathrm{d}_\mathcal{S}) \right\rangle - \upsilon_k \mathrm{KL}\left(\pi(\cdot|s, \mathrm{d}_\mathcal{S}) \| \pi_k(\cdot|s, \mathrm{d}_\mathcal{S})\right) \right\}$$

*yields $\overline{\pi}_{k+1} \propto \exp\{\upsilon_k^{-1} F_{\theta_k}^Q + \tau_k^{-1} F_{\alpha_k}^A\}$.*

To take into account that the representable function of actor network resides in $\mathcal{F}_A$, we update the policy by projecting the energy function of $\overline{\pi}_{k+1}$ back to $\mathcal{F}_A$. Specifically, by denoting $\pi_{k+1} \propto \exp\{\tau_{k+1}^{-1} F_{\alpha_{k+1}}^A\}$, we recover the next actor network $F_{\alpha_{k+1}}^A$ (i.e., energy) by performing the following regression task

$$\alpha_{k+1}^* = \underset{\alpha \in \mathbb{B}(R_\alpha, \alpha_0)}{\operatorname{argmin}} \mathbb{E}_{\widetilde{\sigma}_k} \left\{ F_\alpha^A(s, \mathrm{d}_\mathcal{S}, \overline{a}) - \tau_{k+1}\left[ \upsilon_k^{-1} F_{\theta_k}^Q(s, \mathrm{d}_\mathcal{S}, \overline{a}) + \tau_k^{-1} F_{\alpha_k}^A(s, \mathrm{d}_\mathcal{S}, \overline{a}) \right] \right\}^2, \tag{3.2}$$

where $\widetilde{\sigma}_k = \nu_k \pi_0$. We approximately solve (3.2) via T-step stochastic gradient descent (SGD), and output $\alpha_{k+1} = \alpha(T) \approx \alpha_{k+1}^*$. At the $t$-th iteration of SGD,

$$\alpha(t+1/2) = \alpha(t) - \eta \nabla_\alpha F_{\alpha(t)}^A(s, \mathbf{s}, \overline{a}) \big\{ F_{\alpha(t)}^A(s, \mathbf{s}, a) - \tau_{k+1} \big( v_k^{-1} F_{\theta_k}^Q(s, \mathbf{s}, \overline{a}) + \tau_k^{-1} F_{\alpha_k}^A(s, \mathbf{s}, \overline{a}) \big) \big\},$$

$$\alpha(t+1) = \Pi_{\mathbb{B}(R_\alpha, \alpha_0)} \left( \alpha(t+1/2) \right),$$

where we sample $(s, \mathrm{d}_\mathcal{S}, \overline{a}) \sim \widetilde{\sigma}_k$, and $\mathbf{s} \overset{\text{i.i.d.}}{\sim} \mathrm{d}_\mathcal{S}$, and $\eta$ is the step size. The details are summarized in Algorithm 3 of Appendix A. Finally, we present the complete MF-PPO in Algorithm 1.

---

**Algorithm 1** Mean-Field Neural Proximal Policy Optimization

---

**Require**: Mean-field MDP $(\mathcal{S} \times, \mathcal{P}(\mathcal{S}), \overline{\mathcal{A}}, \mathbb{P}, r)$, discount factor $\gamma$; number of outer iterations $K$, number of inner updates $T$; policy update parameter $v$, step size $\eta$, projection radius $R_\alpha, R_\theta$.
**Initialize:** $\tau_0 \leftarrow 1, F^{A,0} \leftarrow 0, \pi_0 \propto \exp\{\tau_0^{-1} F^{A,0}\}$ (uniform policy).
**for** $k = 0, \ldots, K-1$ **do**
    Set temperature parameter $\tau_{k+1} \leftarrow v\sqrt{K}/(k+1)$, and proximity parameter $v_k \leftarrow v\sqrt{K}$
    Solve (3.1) to update the critic network $F_{\theta_k}^Q$, using TD update (Algorithm 2)
    Solve (3.2) to update the actor network for $F_{\alpha_{k+1}}^A$, using SGD update (Algorithm 3)
    Update policy: $\pi_{k+1} \propto \exp\{\tau_{k+1}^{-1} F_{\alpha_{k+1}}^A\}$
**end for**

---

## 4 GLOBAL CONVERGENCE OF MF-PPO

We present the global convergence of MF-PPO algorithm for the two-layer permutation-invariant parameterization of the actor and critic networks. We remark that our analysis can be extended to multi-layer permutation-invariant networks, and we present the two-layer case here for simplicity of exposition. Specifically, the actor and critic networks take the form

$$F_\alpha^A(s, \mathbf{s}, \overline{a}) = \frac{1}{\sqrt{mN}} \sum_{j=1}^{m_A} \sum_{s' \in \mathbf{s}} u_j \sigma\left(\alpha_j^\top(s, s', \overline{a})\right), \quad F_\theta^Q(s, \mathbf{s}, \overline{a}) = \frac{1}{\sqrt{mN}} \sum_{j=1}^{m_Q} \sum_{s' \in \mathbf{s}} v_j \sigma\left(\theta_j^\top(s, s', \overline{a})\right),$$

where $m_A$ (resp. $m_Q$) is the width of the actor (resp. critic) network, and $\sigma(x) = \max\{x.0\}$ denotes the ReLU activation. We randomly initialize $u_j$ (resp. $v_j$) and first layer weights $\alpha_0 = [\alpha_{0,1}^\top, \ldots, \alpha_{0,m_A}^\top]^\top \in \mathbb{R}^{d \cdot m_A}$ (resp. $\theta_0 \in \mathbb{R}^{d \cdot m_Q}$) by

$$u_j \sim \text{Unif}\{-1, +1\}, \alpha_{0,j} \sim \mathcal{N}(0, \mathrm{I}_d/d), \quad \forall j \in [m].$$

For ease of analysis, we take $m = m_A = m_Q$ and share the initialization of $\alpha_0$ and $\theta_0$ (resp. $u_0$ and $v_0$). Additionally, we keep $u_j$'s fixed during training, and $\alpha$ (resp. $\theta$) within ball $\mathbb{B}(\alpha_0, R_A)$ (resp. $\mathbb{B}(\theta_0, R_Q)$) throughout training. We define the following function class which approximates the class of previously defined actor/critic network for large network width.

**Definition 4.1.** *Given $R_\beta > 0$, define the function class over domain $\mathcal{S} \times \mathcal{S} \times \mathcal{A}$ by*

$$\mathcal{F}_{\beta,m} = \Big\{ f_\beta(\cdot) \Big| f_\beta(s, s', \overline{a}) = \frac{1}{\sqrt{m}} \sum_{j=1}^m v_j \mathbb{1}\{\beta_{0,j}^\top(s, s', \overline{a}) > 0\} \beta_j^\top(s, s', \overline{a}), \|\beta - \beta_0\|_2 \leq R_\beta \Big\},$$

*where $v_j, \beta_{0,j}$ are random weights sampled according to*

$$v_j \sim \text{Unif}\{-1, +1\}, \beta_{0,j} \sim \mathcal{N}(0, \mathrm{I}_d/d), \quad \forall j \in [m].$$

*$\mathcal{F}_{\beta,m}$ also induces the function class over $\mathcal{S} \times \mathcal{P}(\mathcal{S}) \times \mathcal{A}$ given by*

$$\mathcal{F}_{\beta,m}^{\mathcal{P}} = \big\{ F(\cdot) \,\big|\, F(s, \mathrm{d}_\mathcal{S}, \overline{a}) = \mathbb{E}_{s' \sim \mathrm{d}_\mathcal{S}} f_\beta(s, s', \overline{a}), f_\beta \in \mathcal{F}_{\beta,m} \big\}.$$

It is well known that functions within $\mathcal{F}_{\beta,m}$ approximate functions within the reproducing kernel Hilbert space associated with kernel $\mathrm{K}(x, y) = \mathbb{E}_{z \sim \mathcal{N}(0, I_d/d)} \big\{ \mathbb{1}(z^\top x > 0, z^\top y > 0) \big\}$ for a large network width $m$ (Jacot et al., 2018; Chizat and Bach, 2018; Cai et al., 2019; Arora et al., 2019) and whose RKHS norm is bounded by $R_\beta$. For large $R_\beta$ and $m$, $\mathcal{F}_{\beta,m}$ represents a rich class of functions. Additionally, functions within $\mathcal{F}_{\beta,m}^{\mathcal{P}}$ can be viewed as the mean-embedding of the joint state-action pair onto the RKHS space (Muandet et al., 2016; Song et al., 2009; Smola et al., 2007). Below, we make one important assumption, which assumes that $\mathcal{F}_{\beta,m}^{\mathcal{P}}$ is rich enough to represent the Q-function of all the policies within our policy class.

**Assumption 1.** *For any policy $\pi$ induced by $F_A \in \mathcal{F}^A$, we have $Q^\pi \in \mathcal{F}^{\mathcal{P}}_{\theta, m_Q}$.*

We remark that Assumption 1 can be relaxed into requiring that $\mathcal{F}^{\mathcal{P}}_{\beta, m}$ has $\epsilon$ approximation error when parameterizing the set of Q-functions, with an additional $\epsilon$ term appearing in the convergence bound developed in Theorem 4.1 (Lan, 2021).

We define mild conditions stating boundedness of reward, and regularity of stationary distributions.

**Assumption 2.** *Reward function $r(\cdot) \leq \overline{r}$ for some $\overline{r} > 0$. Additionally, there exists $c > 0$ such that $\mathbb{E}\left\{\mathbb{1}\left(|z^\top(s, s', a)| \leq t\right)\right\} \leq c \cdot \frac{t}{\|z\|_2}$ for any $z \in \mathbb{R}^d$ and $t > 0$.*

We measure the progress of MF-PPO in Algorithm 1 using the expected total reward

$$\mathcal{L}(\pi) = \mathbb{E}_{\nu^*}\left[V^\pi(s, \mathrm{d}_{\mathcal{S}})\right] = \mathbb{E}_{\nu^*}\left\{\langle Q^\pi(\cdot|s, \mathrm{d}_{\mathcal{S}}), \pi(\cdot|s, \mathrm{d}_{\mathcal{S}})\rangle\right\}, \tag{4.1}$$

where $\nu^*$ is the stationary state distribution of the optimal policy $\pi^*$. We also denote $\sigma^*$ as the stationary state-action distribution induced by $\pi^*$. Note that we have: $\mathcal{L}(\pi^*) = \mathbb{E}_{\nu^*}\left[V^{\pi^*}(s, \mathrm{d}_{\mathcal{S}})\right] \geq \mathbb{E}_{\nu^*}\left[V^\pi(s, \mathrm{d}_{\mathcal{S}})\right] = \mathcal{L}(\pi)$, for any policy $\pi$. Our main results are presented in the following theorem, showing that $\mathcal{L}(\pi_k)$ converges to $\mathcal{L}(\pi^*)$ at a sub-linear rate.

**Theorem 4.1** (Global Convergence of MF-PPO). *Under Assumptions 1 and 2, the policies $\{\pi_k\}_{k=1}^K$ generated by Algorithm 1 satisify*

$$\min_{0 \leq k \leq K}\{\mathcal{L}(\pi^*) - \mathcal{L}(\pi_k)\} \leq \frac{\upsilon\left(\log|\overline{\mathcal{A}}| + \sum_{k=1}^{K-1}(\varepsilon_k + \varepsilon'_k)\right)}{(1-\gamma)\sqrt{K}} + \frac{M}{(1-\gamma)\upsilon\sqrt{K}},$$

*where $M = \mathbb{E}_{\nu^*}\left\{\max_{\overline{a} \in \overline{\mathcal{A}}}(F^Q_{\theta_0}(s, s', \overline{a}))^2\right\} + 2R^2_A$, $\varepsilon_k$ and $\varepsilon'_k$ are defined in Lemma 4.3. In particular, suppose at each iteration of MF-PPO, we observe $N = \Omega\left(K^3 R^4_A + K R^4_Q\right)$ agents, and the actor/critic network satisfies $m_A = \Omega\left(K^6 R^{10}_A + K^4 R^{10}_A|\overline{\mathcal{A}}|^2\right), m_Q = \Omega\left(K^2 R^{10}_Q\right)$, and $T = \Omega\left(K^3 R^4_A + K R^4_Q\right)$, then we have*

$$\min_{0 \leq k \leq K}\{\mathcal{L}(\pi^*) - \mathcal{L}(\pi_k)\} \leq \frac{\upsilon^2\left(\log|\overline{\mathcal{A}}| + \mathcal{O}(1)\right) + M}{(1-\gamma)\upsilon\sqrt{K}}.$$

Theorem 4.1 states that, given sufficiently many agents and a large enough actor/critic network, MF-PPO attains global optimality at a sublinear rate. Our result shows that when solving the mean-field MDP, having more agents serves as a blessing instead of a curse. In addition, as will be demonstrated in our proof sketch, there exists an inherent phase transition, where the final optimality gap is dominated by statistical error for a small number of agents (first phase); and by optimization error for a large number of agents (second phase).

The complete proof of Theorem 4.1 takes careful analysis on the error from policy evaluation (3.1) and the improvement step (3.2). The analysis on the outer iterations of MF-PPO can be overviewed as approximate mirror descent, which needs to take into account how the evaluation and improvement error interacts. Intuitively, the tuple $(\varepsilon_k, \varepsilon'_k)$ describes the the effect of policy update when using approximate policy evaluation and policy improvement, and will be further clarified in the ensuing discussion. We present here the skeleton of our proof, and defer the technical detail to the appendix.

***Proof Sketch.*** We first establish the convergence of the policy evaluation and improvement step.

**Lemma 4.1** (Policy Evaluation). *Under the same assumptions in Theorem 4.1, let $Q^{\pi_k}$ denote the Q-function of policy $\pi_k$, let $\epsilon_k = \mathbb{E}_{\mathrm{init}, \sigma_k}\left(F^Q_{\overline{\theta}(T)}(\cdot) - Q^{\pi_k}(\cdot)\right)^2$ denote policy evaluation error, where $\overline{\theta}(T)$ is the output of Algorithm 2, we have $\epsilon_k = \mathcal{O}\left(\frac{R^2_Q}{T^{1/2}} + \frac{R^{5/2}_Q}{m^{1/4}_Q} + \frac{R^2_Q}{N^{1/2}} + \frac{R^3_Q}{m^{1/2}_Q}\right).$*

**Lemma 4.2** (Policy Improvement). *Under the same assumptions in Theorem 4.1, let $\epsilon'_{k+1} = \mathbb{E}_{\mathrm{init}, \widetilde{\sigma}_k}\left\{F^A_{\overline{\alpha}(T)}(\cdot) - \tau_{k+1}\left(\upsilon^{-1}_k F^Q_{\theta_k}(\cdot) + \tau^{-1}_k F^A_{\alpha_k}(\cdot)\right)\right\}^2$ denote policy optimization error, where $\overline{\alpha}(T)$ is the output of Algorithm 3, we have $\epsilon'_{k+1} = \mathcal{O}\left(\frac{R^2_A}{T^{1/2}} + \frac{R^{5/2}_A}{m^{1/4}_A} + \frac{R^2_A}{N^{1/2}} + \frac{R^3_A}{m^{1/2}_A}\right).$*

Lemma 4.1 and 4.2 show that despite non-convexity, both policy evaluation and policy improvement steps converge approximately to the global optimal solution. In particular, for both policy evaluation steps and improvement steps, given networks with large width, for a small number of iterations $T$, the optimization error $\mathcal{O}\left(T^{-1/2}\right)$ dominates the optimality gap; for a large number of iterations $T$, the statistical error $\mathcal{O}\left(N^{-1/2}\right)$ dominates the optimality gap.

With Lemma 4.1 and 4.2, we illustrate the main argument for the proof of Theorem 4.1. Let us assume the ideal case when $\epsilon_k = \epsilon'_{k+1} = 0$. Note that for $\epsilon_k = 0$, we obtain the exact $Q$-function of policy $\pi_k$. For $\epsilon'_{k+1} = 0$, we obtain the ideal energy-based updated policy define in Proposition 3.2. That is,

$$\pi_{k+1} = \text{argmax}_\pi \, \mathbb{E}_{\nu_k} \big\{ \, \langle Q^{\pi_k}(s, \mathrm{d}_\mathcal{S}, \cdot), \pi(\cdot|s, \mathrm{d}_\mathcal{S}) \rangle \, \upsilon_k \text{KL} \left( \pi(\cdot|s, \mathrm{d}_\mathcal{S}) \| \pi_k(\cdot|s, \mathrm{d}_\mathcal{S}) \right) \, \big\}. \qquad (4.2)$$

Without function approximation, problem (4.2) can be solved by treating each joint state $(s, \mathrm{d}_\mathcal{S})$ independently, hence one can apply the well known three-point lemma in mirror descent (Chen and Teboulle, 1993) and obtain that, for all $(s, \mathrm{d}_\mathcal{S}) \in \mathcal{S} \times \mathcal{P}(\mathcal{S})$:

$$\langle Q^{\pi_k}(s, \mathrm{d}_\mathcal{S}, \cdot), \pi^*(\cdot|s, \mathrm{d}_\mathcal{S}) - \pi_k(\cdot|s, \mathrm{d}_\mathcal{S}) \rangle$$
$$\leq \upsilon_k \big\{ \text{KL} \left( \pi^*(\cdot|s, \mathrm{d}_\mathcal{S}) \| \pi_k(\cdot|s, \mathrm{d}_\mathcal{S}) \right) - \text{KL} \left( \pi^*(\cdot|s, \mathrm{d}_\mathcal{S}) \| \pi_{k+1}(\cdot|s, \mathrm{d}_\mathcal{S}) \right) - \text{KL} \left( \pi_{k+1}(\cdot|s, \mathrm{d}_\mathcal{S}) \| \pi_k(\cdot|s, \mathrm{d}_\mathcal{S}) \right) \big\}$$
$$+ \langle Q^{\pi_k}(s, \mathrm{d}_\mathcal{S}, \cdot), \pi_{k+1}(\cdot|s, \mathrm{d}_\mathcal{S}) - \pi_k(\cdot|s, \mathrm{d}_\mathcal{S}) \rangle.$$

From Lemma 6.1 in Kakade and Langford (2002), the expectation of the left hand side yields exactly $(1 - \gamma) \{ \mathcal{L}(\pi^*) - \mathcal{L}(\pi_k) \}$. Hence we have

$$(1 - \gamma) \{ \mathcal{L}(\pi^*) - \mathcal{L}(\pi_k) \}$$
$$\leq \upsilon_k \mathbb{E}_{\nu^*} \big\{ \text{KL} \left( \pi^*(\cdot|s, \mathrm{d}_\mathcal{S}) \| \pi_k(\cdot|s, \mathrm{d}_\mathcal{S}) \right) - \text{KL} \left( \pi^*(\cdot|s, \mathrm{d}_\mathcal{S}) \| \pi_{k+1}(\cdot|s, \mathrm{d}_\mathcal{S}) \right)$$
$$- \text{KL} \left( \pi_{k+1}(\cdot|s, \mathrm{d}_\mathcal{S}) \| \pi_k(\cdot|s, \mathrm{d}_\mathcal{S}) \right) \big\} + \mathbb{E}_{\nu^*} \langle Q^{\pi_k}(s, \mathrm{d}_\mathcal{S}, \cdot), \pi_{k+1}(\cdot|s, \mathrm{d}_\mathcal{S}) - \pi_k(\cdot|s, \mathrm{d}_\mathcal{S}) \rangle.$$

Pinsker's Inequality $\text{KL} \left( \pi_{k+1}(\cdot|s, \mathrm{d}_\mathcal{S}) \| \pi_k(\cdot|s, \mathrm{d}_\mathcal{S}) \right) \geq \frac{1}{2} \|\pi_{k+1} - \pi_k\|_1^2$, combined with observation $\|Q^{\pi_k}(s, \mathrm{d}_\mathcal{S}, \cdot)\|_\infty \leq \overline{r}/(1 - \gamma)$, and basic inequality $-ax^2 + bx \leq b^2/(4a)$ for $a > 0$ gives us

$$(1 - \gamma) \{ \mathcal{L}(\pi^*) - \mathcal{L}(\pi_k) \}$$
$$\leq \upsilon_k \mathbb{E}_{\nu^*} \big\{ \text{KL} \left( \pi^*(\cdot|s, \mathrm{d}_\mathcal{S}) \| \pi_k(\cdot|s, \mathrm{d}_\mathcal{S}) \right) - \text{KL} \left( \pi^*(\cdot|s, \mathrm{d}_\mathcal{S}) \| \pi_{k+1}(\cdot|s, \mathrm{d}_\mathcal{S}) \right) \big\} + \frac{\overline{r}^2}{2\upsilon_k(1 - \gamma)^2}.$$

By setting $\upsilon_k = \mathcal{O}(\sqrt{K})$, and telescoping the above inequality from $k = 0$ to $K - 1$, we obtain: $\min_{0 \leq k \leq K-1} \{ \mathcal{L}(\pi^*) - \mathcal{L}(\pi_k) \} = \mathcal{O}(1/\sqrt{K})$. Note that the key element in the global convergence of MF-PPO is the recursion defined in the previous inequality, which holds whenever we have an exact $Q$-function of the current policy and no function approximation is used when updating the next policy. Now MF-PPO conducts approximate policy evaluation $\epsilon_k > 0$, and after obtaining the approximate $Q$-function, conducts approximate policy improvement step $\epsilon'_{k+1} > 0$ with function approximation. In addition, the error of approximating the $Q$-function introduced in the evaluation step can be further compounded in the improvement step. Nevertheless, the previous inequality still holds approximately, with additional terms representing the policy evaluation/improvement errors.

**Lemma 4.3** (Liu et al. (2019a))**.** *Let $\epsilon_k$ (evaluation error) and $\epsilon'_{k+1}$ (improvement error) be defined as in Lemma 4.1 and Lemma 4.2, respectively. We have:*

$$(1 - \gamma) \left( \mathcal{L}(\pi^*) - \mathcal{L}(\pi_k) \right) \leq \upsilon_k \mathbb{E}_{\nu_*} \big\{ \text{KL} \left( \pi^*(\cdot|s, \mathrm{d}_\mathcal{S}) \| \pi_k(\cdot|s, \mathrm{d}_\mathcal{S}) \right) - \text{KL} \left( \pi^*(\cdot|s, \mathrm{d}_\mathcal{S}) \| \pi_{k+1}(\cdot|s, \mathrm{d}_\mathcal{S}) \right) \big\}$$
$$+ \upsilon_k \left( \varepsilon_k + \varepsilon'_k \right) + \upsilon_k^{-1} M. \qquad (4.3)$$

*where*

$$\varepsilon_k = \tau_{k+1}^{-1} \epsilon'_{k+1} \phi^*_{k+1} + \upsilon_k^{-1} \epsilon_k \psi^*_k, \quad \varepsilon'_k = |\mathcal{A}| \tau_{k+1}^{-2} (\epsilon'_{k+1})^2, \quad M = \mathbb{E}_{\nu^*} \left\{ \max_{\overline{a} \in \overline{\mathcal{A}}} \left[ F_{\theta_0}^Q(s, \mathrm{d}_\mathcal{S}, \overline{a}) \right]^2 \right\} + 2R_A^2.$$

*In addition, $\phi^*_k$ and $\psi^*_k$ are defined by:*

$$\phi^*_k = \mathbb{E}_{\widetilde{\sigma}_k} \left[ |\mathrm{d}\pi^*/\mathrm{d}\pi_0 - \mathrm{d}\pi_k/\mathrm{d}\pi_0|^2 \right]^{1/2}, \quad \psi^*_k = \mathbb{E}_{\sigma_k} \left[ |\mathrm{d}\sigma^*/\mathrm{d}\sigma_k - \mathrm{d}(\nu^* \times \pi_k)/\mathrm{d}\sigma_k|^2 \right]^{1/2}.$$

Finally, by telescoping inequality (4.3) from $k = 0$ to $K - 1$, we complete the proof of Theorem 4.1.

## 5 EXPERIMENTS

We perform experiments on the benchmark multi-agent particle environment (MPE) used in prior work (Lowe et al., 2017; Mordatch and Abbeel, 2018; Liu et al., 2019b). In the *cooperative navigation* task, $N$ agents each with position $x_i \in \mathbb{R}^2$ must move to cover $N$ fixed landmarks at positions $y_i \in \mathbb{R}^2$. They receive a team reward $R = - \sum_{i=1}^N \min_{j \in [N]} \|y_i - x_j\|_2$; In the *cooperative push* task, $N$ agents with position $x_i \in \mathbb{R}^2$ work together to push a ball $x \in \mathbb{R}^2$ to a fixed landmark $y \in \mathbb{R}^2$. They receive a team reward $R = - \min_{j \in [N]} \|x_j - x\|_2 - \|x - y\|_2$. Both tasks involve homogeneous

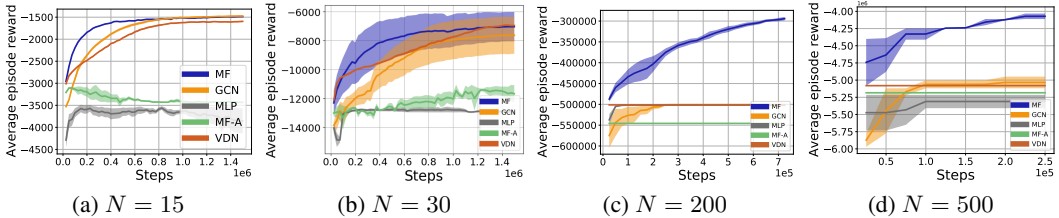

(a) $N = 15$     (b) $N = 30$     (c) $N = 200$     (d) $N = 500$

Figure 3: Performance versus number of environment steps in the multi-agent *cooperative navigation* task, over five independent runs per method. Points are taken every 1000 training episodes, with the first point taken after the first 1000, and is the average reward of 1000 evaluation episodes. MF significantly outperforms other critic representations for various number of agents.

agents, and all the agents share the same team reward. Note that MPE environment also models interaction between agents, including collision, and the collided agents receive negative rewards.

We instantiate our method **MF**, by parameterizing the centralized critic function using a DeepSet (Zaheer et al., 2017) network, with two hidden layers. We use a standard two-layer multi-layer perception (MLP) for the centralized actor network in all algorithms. The actor network outputs the mean and diagonal covariance of a Gaussian distribution over the joint action space. We refer interested readers to Appendix B for detailed configurations of hyperparameters.

We compare with two other critic representations: one that uses MLP for the centralized critic, labeled **MLP**, and another that uses a graph convolutional network for the critic (Liu et al., 2019b), labeled **GCN**. Note that the GCN representation is permutation invariant if one imposes a fully-connected graph for the agents in the MPE, but this invariance property does not hold for all graphs in general. We also compare with an extension of (Yang et al., 2018) to the case of continuous action spaces, labeled **MF-A**, in which each independent

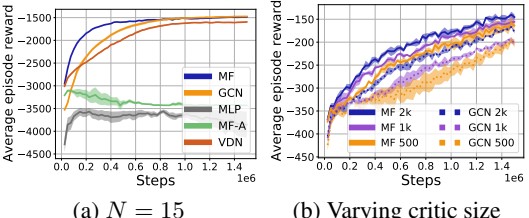

(a) $N = 15$     (b) Varying critic size

Figure 4: (a) Performance versus number of environment steps in the multi-agent *cooperative push* task. (b) MF outperforms GCN even with a fewer number of critic network parameters ($N = 3$).

DDPG agent $i$ has a decentralized critic $Q(s_i, a_i, \bar{a}_i)$ that takes in the mean of all other agents' actions $\bar{a}_i := \frac{1}{N-1} \sum_{j \neq i} a_j$. Finally, we include comparison with **VDN** (Sunehag et al., 2017), where the centralized critic network is the direct summation of local critic networks and thus being permutation invariant. Empirically, as we find that off-policy RL learns faster than on-policy RL in the MPE with higher agent number, regardless of the critic representation, we make all comparisons on top of MADDPG (Lowe et al., 2017). For a fair comparison of all critic representations, we ensure that all neural network architectures contain approximately the same number of trainable weights.

For the cooperative navigation task, Figure 3 shows that the permutation invariant critic representation based on DeepSet enables MF to learn faster or reach a higher performance than all other representations and methods in the MPE with 15, 200, and 500 agents. For the cooperative push task, Figure 4a demonstrates a similar performance improvement provided by MF. In addition, we also demonstrate the superior parameter efficiency of MF compared to GCN. Figure 4b shows that MF consistently and significantly outperforms GCN as the number of parameters in their critic network varies over a range, with all other settings fixed. In particular, MF requires much fewer critic parameters to achieve higher performance than GCN.

**Computational Improvements**. Theorem 4.1 states that to obtain a small optimality gap in MF-PPO, one needs to compute the update on a large number of agents. We remark that with the dual embedding techniques developed in Dai et al. (2017), one can avoid computation on all the agents by sampling a small number of agents to compute the update. This technique could be readily incorporated into MF-PPO to improve its computational efficiency.

**Conclusion**. We propose a principled approach to exploit agent homogeneity and permutation invariance through the mean-field approximation in MARL. Our results are also the first to show the global convergence of MARL algorithms with neural networks as function approximators. This is in sharp contrast to current practices, which are mostly heuristic methods without convergence guarantees.

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
