# OpenReview forum: "A Principled Permutation Invariant Approach to Mean-Field Multi-Agent Reinforcement Learning"
_ICLR.cc/2022/Conference — ICLR 2022 Submitted_

### Official Review · Reviewer_979E · 2021-11-01

**Correctness:** 4
**Technical Novelty And Significance:** 3
**Empirical Novelty And Significance:** 3
**Recommendation:** 8
**Confidence:** 5

**Main Review:**

The paper is nicely written and easy to follow. I like very much the way that the authors motivate and demonstrate how the permutation invariance property helps to reduce the complexity of the MARL problem with homogeneous agents.

The techniques used to show the sample complexity is based on (1) the neural policy gradient paper (Wang, Cai, Yang, and Wang, 2019) for the single-agent case and (2) the law of large number used in the mean-field approximation. I am confident that the main results are correct.

I have several minor suggestions that may help to further improve the exposition:

(1) The paragraph starting with "To scale MARL algorithms ..." on page 2: The authors gave an overview on mean-field MARL in this paragraph. I view mean-field game (MFG) and mean-field control (MFC) are two different sets of problems with the MFG solving for Nash equilibria and the MFC solving for social optimal solutions. The authors did not distinguish these two when providing the overview. I suggest the authors divide the literature review into two parts and treat the references for MFG and MFC separately. A reference that might be relevant to include: Mean-Field Multi-Agent Reinforcement Learning: A Decentralized Network Approach (Gu et al., 2021)

(2) Permutation mapping $\kappa(\cdot)$: I understand what the authors mean by "permutation mapping" and it can be any function of the empirical distribution (e.g, first moment and second moment). But I suggest the authors provide a more rigorous (mathematical) definition of permutation mapping.

(3) Factorization $\nu(a|s)$ on page 3: The authors may add more discussions on $o_i$ which is the informaton observed by agent $i$.

(4) Generality of the definition for mean-field MDP: Is it possible to include $(a,d_{\mathcal{A}})$ in the formulation (reward and transition) instead of only the average action ($\bar{a}$)?

(5) Theorem 4.1: The authors may add a discussion on the comparison to the single-agent paper (Wang, Cai, Yang, and Wang, 2019) and highlight the additional technical differences (i.e., handling the mean-field part). How does the term $M$ depend on N?

(6) Lemma 4.1: The authors may add a discussion on why $\sqrt{1/N}$ appears in the upper bound (I believe it's from the law of large numbers).



**Summary Of The Paper:**

This paper proposes a mean-field proximal policy optimization algorithm (MF-PPO) for MARL with a large population of homogenuous agents. The sample complexity is derived based on a two-layer neural network approximation and the proposed algorithm is tested in several detailed experiments.

**Summary Of The Review:**

The submission is well written and provides a good contribution to the MARL literature.

---

> ### Author Response · Authors · 2021-11-17
> **Response to Reviewer 979E**
>
> We deeply appreciate your positive feedbacks and constructive suggestions on improving the paper. Below we provide our responses to your comments in detail.
>
> 1. We thank the reviewer for pointing out the distinction between MFG and MFC and the pointer to the related reference.
> We have made separate discussions on these two topics, including the related reference in our updated literature review.
>
> 2. We thank the reviewer for the suggestion on making our presentation more rigorous. We have included the corresponding definitions of the permutation mapping in the updated draft. Kindly refer to the updated definition after equation (2.1).
>
> 3. Thanks for pointing out this valuable suggestion.
>  For clarity, we remark here that the agent is able to observe its local state $s$, and the aggregate global state information (presented in $\hat{d}_\mathcal{S}$). Kindly refer to our updated discussions under Definition 2.1 on our modeling process.
>
> 4. Thanks for the valuable suggestion and this is indeed a great question we are still exploring. Note that our current development shows logarithmic dependence on action space $\overline{A}$, which is finite. When considering $(a, d_{\mathcal{A}})$ as the mean-field action, the action space is infinite, which is something out of our current analysis framework. We leave this direction of investigation as an important open question.
>
> 5. Thanks for the valuable suggestion on the discussion with a single-agent algorithm, which we will include in our updated version.
> We gently remark that the term $M$ is independent of the number of agents $N$, as it is not related to the estimation of the unknown mean-field state.
>
> 6. Thanks for the valuable suggestion on the supplementary discussion.
> The dependence of $1/\sqrt{N}$ exactly comes from estimating the mean-field state using finite samples (i.e., law of large numbers).
> We will make corresponding discussions in the updated version.

---

### Official Review · Reviewer_6Xxz · 2021-11-02

**Correctness:** 3
**Technical Novelty And Significance:** 3
**Empirical Novelty And Significance:** 2
**Recommendation:** 5
**Confidence:** 2

**Main Review:**

I am shared about this paper. It seems to be a valid theoretical contribution, but since it is beyond the scope of my expertise, it is hard for me to fully validate/appreciate. As to the experimental part, I find it to be relatively weak, which might still be ok, if the theoretical part is strong enough.

I raise some concrete questions, which hopefully will clarify some of my doubts during the rebuttal.

1. Why are energy models used for policy training, is it matter of convenience, or is there a deeper reason?

2. Why there is restriction to $B(\theta_0, R_\theta)$? Shouldn't it be $B(\theta_{k-1}, R_\theta)$ in the equation (3.1).

3. Why the mean-field MDP uses $(s, \text{d}_\mathcal{S})$. It should be enough to just take  $\text{d}_\mathcal{S}$ (i.e. none of the agents is special)? [The same question applies to Prop 2.2].

4. Is the proof a novel one?

5. What is the exact experimental setup:

6. 1. What are the details about the environments used? What are the interactions between agents? How intensive are they?
   2. What is the algorithmic setup? I am confused as it suggests using DDPG, which is somewhat detached from the theoretical analysis.

**Summary Of The Paper:**

The authors present a principled method of solving problems with multiple homogenous agents.

**Summary Of The Review:**

As for the moment, I find the contribution below the standard of a top-tier conference like ICLR. It is not clear what the exact theoretical contributions are and how they are related to the experimental part.

---

> ### Author Response · Authors · 2021-11-17
> **Response to Reviewer 6Xxz, Part I**
>
> We deeply appreciate your reviews and detailed comments. Below we provide our response to your detailed remarks and hope it can address your current reservations.
>
> 1. We remark that the energy-based model adapts better to the PPO optimization. As discussed in Proposition 3.2 of our work, if the policy $\pi_k$ is energy-based and the policy optimization step updates $\pi_{k+1}$ by solving the PPO objective, then the exact solution of $\pi_{k+1}$ is also an energy-based policy.
> In our implementation, the policy is also parameterized as an energy-based policy. In addition, we remark that such exact-form updates play a central role in establishing the convergence results of our paper.
>
> 2. We introduce the restriction to adapt to the analysis of neural networks based on the neural tangent kernel (NTK) framework. Intuitively, we require that the parameter update should not deviate far away from the initialization. In addition, we remark that such restriction can be further relaxed based on recent advances in the analysis of neural networks. See, e.g., [1] for the discussion.
>
>
>
> 3. The reason why we introduce $(s, \mathrm{d}_{\mathcal{S}})$ as the joint space is to model the problem from the perspective of any agent.
> We briefly reiterate our modeling process here to provide more context for understanding.
> Since the agents are homogeneous, the system is the same from any agent's perspective.
> We choose any agent (referred to as the representative agent), the state information of such an agent includes the local state $s$, and the mean-field state $d_\{\mathcal{S}\}$.
> With such an information, the agent selects a meta action $\overline{a} \in \overline{\mathcal{A}} \in \mathcal{A}^{\mathcal{S}}$, and uses such a meta action to make local decision $a = \overline{a}(s)$.
> We remark that such modeling of the decision process allows the agent to make decisions both on its local information (local state $s$) and the global information (mean-field state $d_\{\mathcal{S}\}$), which can not be captured by simply modeling $d_\{\mathcal{S}\}$ as the joint state.
> From homogeneity, we assume all the rest of the agents use the same meta action $\overline{a}$ to make their local actions.
> Note that different agents can still make different local actions due to their different local states, i.e., $\overline{a}(z) \neq \overline{a}(z')$ in general for $z \neq z' \in \mathcal{S}$.
>  The joint state at the next timestep $(s', d'_\{\mathcal{S}\})$ naturally depends on the current global state $(s, d_\{\mathcal{S}\})$ and the meta action $\overline{a}$, and is specified by the transition kernel $P(s', d'_\{\mathcal{S}\} | s, d_\{\mathcal{S}\}, \overline{a})$.
>  In addition, the representative agents receives a reward $r(s, d_\{\mathcal{S}\}, \overline{a})$, which depends on the local state and mean-field sate, and the meta action $\overline{a}$ (since all the other agents use $\overline{a}$ to generate their local actions).
>  We have included related discussions on our modeling process in our updated draft, kindly refer to the discussion under Definition 2.1.
>
> 4. We remark that our proof technique is inspired by the prior work on single-agent trust region policy optimization algorithm [2]. Our results, however, require developing techniques that can bound the error of finite sample estimation of the unknown mean-field state $(s, \mathrm{d}_{\mathcal{S}})$ throughout the optimization process, which is completely new in our developments.

---

> > ### Author Response · Authors · 2021-11-17
> > **Response to Reviewer 6Xxz, Part II**
> >
> > 5. As described in Section 5, our experiments were conducted in the multi-agent particle environment, which is a widely-accepted benchmark listed under OpenAI’s open-source repository (https://github.com/openai/multiagent-particle-envs). It has been used in a large number of publications in MARL since 2017.
> >
> >
> > * (5a).This 2D environment models the kinematics and forces of particles with mass and density that take up physical space. Some of these particles are agents under our control, while others are movable objects (e.g. in the cooperative push task) or fixed target landmarks that agents have to reach (e.g. in the cooperative navigation task). Interactions between agent particles and environment objects include contact forces and damping factors during collisions. Regarding the “intensity” of interactions, which represents the difficulty of the problem, we point out that the density of agents per unit area is almost 0.1, meaning that 10\% of the available space on the 2D map is already occupied by agent particles that take up physical space. Note that the number of landmarks is equal to the number of agents in the cooperative navigation task, so almost 20\% of the available space is already covered by objects. Now, further note the fact that agents and landmarks are initialized randomly, so agents need to maneuver around one another to cover landmarks while avoiding collisions that incur negative reward. This makes agent interactions a challenging problem, especially as the number of agents and targets increases. The importance of solving this multi-agent challenge is also indicated by the significant improvement in performance of our approach compared to baselines that we see in Figures 3, 4, and 5.
> >
> > * (5b). As described in the first paragraph on page 9, the practical implementation of MF-PPO is based on using a DeepSet architecture as the model of a centralized critic. This centralized critic can be used in either a MADDPG algorithm or a PPO algorithm. Using it in a PPO algorithm is more aligned with the main theoretical formulation of our paper, but using it in a MADDPG algorithm allows us to make a more fair comparison to Liu et al. 2019b who used a Graph Convolutional Network to represent the critic. We clarify that our paper never says we applied our approach to the DDPG algorithm: the only mention of DDPG is on Page 9 where we described a different baseline method that uses independent DDPG (Yang et al. 2018), which is different from MADDPG and which does not involve a DeepSet-based critic. For more details on the practical implementation, we refer the reviewer to Appendix B.1.
> >
> >
> > **References:**
> >
> > [1] Cayci, Semih, et al. "Sample Complexity and Overparameterization Bounds for Temporal Difference Learning with Neural Network Approximation." arXiv preprint arXiv:2103.01391 (2021).
> >
> > [2] Liu, Boyi, et al. "Neural proximal/trust region policy optimization attains globally optimal policy." arXiv preprint arXiv:1906.10306 (2019).

---

### Official Review · Reviewer_fcc8 · 2021-11-04

**Correctness:** 2
**Technical Novelty And Significance:** 2
**Empirical Novelty And Significance:** 3
**Recommendation:** 3
**Confidence:** 4

**Main Review:**

Multi-agent cooperative systems are important research topics and using mean-field approximation to design algorithms is an interesting research direction. This paper gives some theoretical analysis on the motivation of the mean-field approximation and proposes algorithm MF-PPO to solve mean-field MDP, which is new in related fields. However, there are also some major issues, as listed below.
1. This paper provide some analysis on multi-agent cooperative systems with permutation invariance property, to motivate mean-field MDP. However, I have some concerns with respect to the motivation.
    * In proposition 2.2, function $g_\nu$ should have some conditions: for example, it is possible that it depends on the number of agents $N$ in the system (when $r=\sum_{i=1}^N s_i$, it is permutation invariant and $g_\nu$ depends on $N$). In such cases, it’s not clear to see why the corresponding mean-field MDP exists when $N$ goes to infinity. There seems to be some additional assumptions needed other than homogeneity and permutation invariance. Besides, the proof of Proposition 2.2 is hard to follow. The authors should make it clearer why Theorem 11 in [Bloem-Reddy and Teh (2019)] can be adapted.
    * In the finite-agent MDP, the policies considered are randomized policies (which I infer from the notation $a_t\sim \nu(s_t)$). However in the mean-field MDP, the policy for each agent $\bar a$ becomes deterministic (as it says $\bar a:\mathcal{S}\rightarrow\mathcal{A}$). The authors should give some explanations here why the space of the agent’s policies changes. In sum, it would be nice if the authors can provide more explanations on why Definition 2.1 can be viewed as the corresponding limit model, given that they claim “the mean-field MDP has a step-to-step correspondence with the finite-agent MDP with homogeneous agents”.
2. Proposition 3.1 seems standard for any symmetric game. See for example [A]. This does not rely on mean-field approximation or the actor critic framework. The benefit of using mean-field approximation is not clear, and the authors should elaborate more on that. Especially the permutation invariance idea for MARL has already been explored in [Liu et al. (2019b)].
3. Given that the authors already cited [Gu et al. (2019), Gu et al. (2020)] and they also proposed mean-field MARL algorithms, it is not clear why they are not compared in the numerical experiments.
4. Some notations in the paper are not rigorous. For example, the space of $\nu$ and $\pi$ are not specified. On top of page 5, $\sigma_k=\nu_k\pi_k$ seems problematic.

[A] Computing equilibria in multi-player games. Christos H. Papadimitriou and Tim Roughgarden.

**Summary Of The Paper:**

This paper deals with a class of cooperative MARL problems with permutation invariance. It first shows that, for such problems, there exists an optimal policy that is permutation invariant, and the value function can be characterized as a function of the local state of one agent and the empirical state distribution over the rest of agents. Based on these observations, the authors introduce the mean-field MDP as the limit of the MARL problem with infinitely many homogeneous agents and design a mean-field proximal policy optimization (MF-PPO) algorithm to solve it. It shows with permutation invariance, the search space of the actor/critic network polynomially depends on the number of agents $N$ and establishes the global convergence of MF-PPO. Some numerical results show better performance compared with some existing algorithms.

**Summary Of The Review:**

This paper proposes to combine mean-field approximation with the permutation invariance idea in [Liu et al., 2019b], which is an interesting direction. Both theoretical analyses and numerical experiments are provided. However, the theoretical justification of the mean-field approximation is very unclear. Also, some closely related algorithms are not compared.

---

> ### Author Response · Authors · 2021-11-17
> **Response to Reviewer fcc8**
>
> We deeply appreciate your valuable comments and constructive feedback. Below we provide our responses to your detailed remarks, which we hope can address your current reservations.
>
> 1a. We agree that function $g$, in general, can implicitly depend on the number of agents $N$. Here we outline our proof of Proposition 2.2. in more detail.
> Note that from the permutation invariance, we can show $ \sum_\{t\geq 0\} \gamma^t r(s^t, \mathbf{s}^t_r, \overline{a}^t)|_\{\mathbf{s}^0_r = \mathbf{s}\}  \overset{d}{=} \sum_\{t \geq 0\} \gamma^t  r(s^t, \mathbf{s}^t_r, \overline{a}^t) |_\{\mathbf{s}^0_r = \kappa(\mathbf{s})\}$ for any permutation mapping $\kappa$ of order $N$.
> Hence from Theorem 11 of [1], we know that there exists a function $g$ such that
> $\sum_\{t\geq 0\} \gamma^t r(s^t, \mathbf{s}^t_r)|_\{\mathbf{s}^0_r = \mathbf{s}\}  \overset{d}{=} g(s, \eta, \hat{p}_\{\mathbf{s}_r\})$.
> Hence we conclude with $V^\nu (s, \mathbf{s}_r) = E_\eta g(s, \eta, \hat{p}_\{\mathbf{s}_r\})$.
> Kindly refer to the updated appendix for our revised presentation for the proof of Proposition 2.2.
>
> [1] Bloem-Reddy, Benjamin, and Yee Whye Teh. "Probabilistic Symmetries and Invariant Neural Networks." J. Mach. Learn. Res. 21 (2020): 90-1.
>
> 1b. We remark that in the mean-field MDP, the action space is the lifted action space, in the sense that the $\overline{a}$ is a deterministic mapping from $\mathcal{S}$ to $\mathcal{A}$ (i.e., a deterministic policy).
> We briefly reiterate our modeling process here for a bit more context to facilitate understanding.
> Since the agents are homogeneous, the system is the same from any agent's perspective.
> We choose any agent (referred to as the representative agent), the state information of such an agent includes the local state $s$, and the mean-field state $d_\{\mathcal{S}\}$.
> With such state information, the agent selects a meta action $\overline{a} \in \overline{\mathcal{A}} \in \mathcal{A}^{\mathcal{S}}$, and uses such a meta action to make local decision $a = \overline{a}(s)$.
> We remark that such modeling of the decision process allows the agent to make decisions both on its local information (local state $s$) and the global information (mean-field state $d_\{\mathcal{S}\}$).
> From homogeneity, we assume all the rest of the agents use the same meta action $\overline{a}$ to make their local actions.
> Note that different agents can still make different local actions due to their different local states, i.e., $\overline{a}(z) \neq \overline{a}(z')$ in general for $z \neq z' \in \mathcal{S}$.
>  The joint state at the next timestep $(s', d'_\{\mathcal{S}\})$ naturally depends on the current global state $(s, d_\{\mathcal{S}\})$ and the meta action $\overline{a}$, and is specified by the transition kernel $P(s', d'_\{\mathcal{S}\} | s, d_\{\mathcal{S}\}, \overline{a})$.
>  In addition, the representative agents receives a reward $r(s, d_\{\mathcal{S}\}, \overline{a})$, which depends on the local state and mean-field sate, and the meta action $\overline{a}$ (since all the other agents use $\overline{a}$ to generate their local actions).
>  Kindly refer to our updated discussion under Definition 2.1 for our included discussion aforementioned modeling process.
>
> 2. We gently remark that our claim of Proposition 3.1 indeed does not reply on the game structure. Indeed, we merely discuss the search space of a permutation-invariant Q-function.
> The reason why we impose the permutation invariance structure of Q-function is presented precisely in Proposition 2.1. Namely, the Q-function is permutation invariant given homogeneous agents and permutation invariant policy.
> The mean-field approximation arises from the structure observation presented in Proposition 2.2.
> Namely, the value function should be a function of the empirical distribution over joint states.
> We remark that although the idea of permutation invariance has been explored in Liu et al. (2019 b).
> The proposed method therein is heuristic-based, without rigorous justification on why permutation invariance structure is a good network inductive bias for homogeneous systems.
>
>
> 3. We remark that the algorithms proposed in Gu et al. (2019), Gu et al. (2020) require discretizing the mean of states and actions by a two-dimensional grid and further conducting kernel-based Q-learning on the grid, and there are no openly available implementations. When such a grid is dense (high precision discretization), the algorithms become computationally inefficient (note that the size of grid grows exponentially w.r.t. the discretization precision). In contrast, we remark that our proposed MF-PPO and the compared baselines scale well when the state spaces and number of agents are large.
> Thus, we do not include the algorithms proposed therein in our comparison.
>
> 4. Thanks for pointing out the unexplained notations. We have included the definition of policy $\nu$ (top of page 3), and $\pi$ (top of page 4), along with clarified notation of $\sigma_k$ in our updated draft.

---

### Decision · Program_Chairs · 2022-01-20

**Decision:**

Reject

**Comment:**

This paper proposes a new multi-agent RL algorithm, based on the PPO algorithm, that uses a mean-field approximation, which results in a a permutation- invariant actor-critic neural architecture. The paper includes a detailed theoretical analysis that shows that the algorithm finds a globally optimal policy at a sub-linear rate of convergence, and that its sample complexity is independent of the number of agents. The paper include some experiments that validate the proposed algorithm.

The reviews of this paper are mixed. Most of the reviewers appreciate the theoretical analysis, but one reviewer does not find the theoretical justification of the mean-field approximation clear. The reviewer also points out to the absence of comparisons to relevant competing algorithms. These concerns are addressed by the authors in their rebuttal. A key issue with this work is the weakness of the empirical evaluation. The proposed method is tested on only two simple tasks, and the results on the second task do not show a considerable advantage of the proposed algorithm. This paper can be strengthened by adding experiments that clearly indicate the advantage of the proposed technique.